# Influence of a Thermo-Mechanical Treatment on the Fatigue Lifetime and Crack Initiation Behavior of a Quenched and Tempered Steel

**Amin Khayatzadeh** [1], **Jan Sippel** [2], **Stefan Guth** [1], **Karl-Heinz Lang** [1] **and Eberhard Kerscher** [2,*]

1  Institute for Applied Materials (IAM-WK), Department of Mechanical Engineering, Karlsruhe Institute of Technology, Engelbert-Arnold-Str. 4, 76131 Karlsruhe, Germany; amin.khayatzadeh@kit.edu (A.K.); stefan.guth@kit.edu (S.G.); khlang@kabelbw.de (K.-H.L.)
2  Materials Testing, Department of Mechanical and Process Engineering, University of Kaiserslautern, Gottlieb-Daimler-Str., 67663 Kaiserslautern, Germany; sippel@mv.uni-kl.de
*  Correspondence: kerscher@mv.uni-kl.de; Tel.: +49-631-205-2136

**Abstract:** A thermo-mechanical treatment (TMT) at the temperature of maximum dynamic strain aging has been optimized and performed on quenched and tempered steel SAE4140H (German designation: 42CrMo4) in order to improve the fatigue limit in the high cycle fatigue (HCF) and and very high cycle fatigue (VHCF) regimes. Fatigue tests, with ultimate cycle numbers of $10^7$ and $10^9$, have shown that the TMT can increase both the fatigue lifetime and the fatigue limit in the HCF and VHCF regimes. The increased stress intensity factors of the critical inclusions after the TMT indicate that the effect can be attributed to a stabilized microstructure around critical crack-initiating inclusions through the locking of edge dislocations by carbon atoms during the TMT.

**Keywords:** high-strength steel; thermo-mechanical treatment (TMT); dynamic strain aging (DSA); high cycle fatigue (HCF); very high cycle fatigue (VHCF); fine granular area (FGA); crack initiation; inclusion

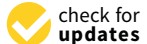

## 1. Introduction

The fatigue limit of high-strength steels is one of the most important mechanical properties for engineering applications, and the demand for steels with a high fatigue limit is ever-increasing [1]. Because of their excellent fatigue properties, quenched and tempered steels are often applied to cyclically loaded components. In the high cycle fatigue (HCF) regime for up to $10^7$ cycles and in the very high cycle fatigue (VHCF) regime for any higher cycle numbers, the fatigue limit of quenched and tempered steels is limited due to inhomogeneities in the microstructure and especially non-metallic inclusions inside the volume from which fatigue cracks initiate and grow [2–5]. The non-metallic inclusions in steels are a result of the physical and chemical processes during steel production [4]. For crack initiation inside the volume, a characteristic smooth fracture surface area, the so-called fisheye, is typically found around the inclusion [5,6]. Particularly for failures after a high number of cycles, a fine granular area (FGA) may be observed in the center of the fisheye around the critical inclusion [6,7]. Many studies have shown that the formation of the FGA plays a significant role in crack initiation and eventual fracture [8–12]. However, the forming mechanism of the FGA is not fully understood [13]. To reach higher fatigue strength in terms of sustainability, for instance, it is of high interest to minimize the negative influence of non-metallic inclusions. One approach is to improve the steel purity and, thus, control the formation of non-metallic inclusions during production [14–16] or to control the inclusions' size with the addition of rare earths [17]. Another possibility is a thermo-mechanical treatment (TMT) in the temperature range where dynamic strain aging (DSA) occurs [18,19]. During DSA, dislocations interact with the alloying element atoms (e.g., carbon in the case of steels) and are, thus, impeded in their movement. Deformation of a material in the

DSA range can lead to a stronger dislocation structure by gathering carbon atoms in the dilatation region of the dislocations, which may cause the generation of a microstructure with a higher fatigue strength [18]. Increasing the fatigue strength of high-strength steels by a TMT has been already shown for the bearing steel SAE52100 (German designation: 100Cr6) in the HCF regime. The results show that a TMT with a cyclic mechanical loading and a gradual increase in stress amplitudes at the temperature of the maximum DSA have the most favorable effect on the fatigue strength [19]. The presumable reason for this is that an increasing load strengthens the microstructure around the inclusions gradually, while a high initial load may directly induce damage around the inclusions [18–20]. The objective of this work is to find a suitable TMT at a temperature of significant DSA for the quenched and tempered steel SAE4140H (German designation: 42CrMo4) in order to improve its fatigue limit and fatigue lifetimes in the HCF and VHCF regimes. The study focuses on the role of inclusions and FGAs for the fatigue and lifetime behavior of specimens with and without the TMT. The fracture surfaces and deleterious defects are investigated using scanning electron microscopy (SEM) and energy-dispersive X-ray spectroscopy (EDX) to analyze their size as well as their chemical composition, respectively.

## 2. Materials and Experimental Procedure

### 2.1. Material and Specimens

The investigated material was the low-alloy steel SAE4140, which exhibits a high fatigue strength after the quenching and tempering heat treatment and is, therefore, typically used for gears and shafts. The chemical composition of the test material is provided in Table 1. The material was delivered in a soft-annealed state, in the form of round bars, from which a near-net-shape geometry of the specimens was machined by turning. These near-net-shape specimens were then heat-treated in a vacuum furnace (ALD Vacuum Technologies GmbH, Hanau, Germany). The heat treatment (HT) consisted of austenitization at 840 °C for roughly 20 min as well as oil quenching to reach room temperature and final tempering at 180 °C for 2 h. After the heat treatment, a fully martensitic microstructure was obtained, which exhibits a hardness of 645 ± 12 HV 0.5. After the quenching and tempering, the final specimen geometries were machined. Since the applied servo-hydraulic test machine for tests in the HCF regime and the ultrasonic test machine for tests in the VHCF regime require different specimen mountings, two specimen geometries were used, which can be viewed in Figure 1. In order to avoid a possible size effect of the specimens, which was found in other studies [21–23], the critically stressed volume of the two specimen geometries was designed to be identical with a cylindrical gauge length of 5 mm and a gauge length diameter of 4 mm. After the initial heat treatment, some of the specimens were thermo-mechanically treated. These are designated as the TMT specimens, while the specimens after the initial heat treatment served as a reference in the fatigue tests and are designated as the HT specimens.

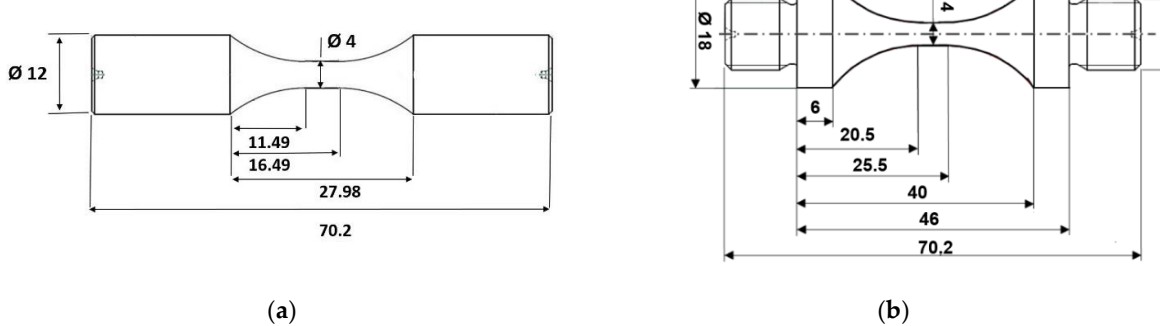

**(a)** **(b)**

**Figure 1.** Specimen geometry used for the experiments (all dimensions in mm): (**a**) for HCF specimens; (**b**) for VHCF specimens.

**Table 1.** Chemical composition of the test material in wt.%.

| C | Si | Mn | P | S | Cr | Mo | Fe |
|---|---|---|---|---|---|---|---|
| 0.430 | 0.259 | 0.743 | 0.012 | 0.039 | 1.060 | 0.207 | Balance |

### 2.2. Testing Setups

All thermo-mechanical treatments, as well as the fatigue tests in the low cycle fatigue (LCF) and HCF regimes, were conducted on a servo-hydraulic push-pull testing machine (Instron GmbH, Darmstadt, Germany) with a capacity of 100 kN. The force was measured with a 100 kN force transducer (Interface Inc., Scottsdale, AZ, USA). The prolongation of the HCF specimens in the experiments to find the temperature of the maximum DSA effects was measured with an extensometer that was applied outside of the specimen's gauge length using alumina rods. The measuring length of the extensometer was 30 mm. With the measured prolongation of the specimen, the actual strain within the cylindrical gauge length cannot be determined since the radii towards the specimen's shoulders are within the measuring length. However, with the applied setup, a qualitative strain in the measuring length of the extensometer can be determined, which is, in turn, used to obtain stress–strain hysteresis loops and a qualitative plastic strain amplitude $\varepsilon_{a,p,quali}$. It is assumed that with this qualitative plastic strain amplitude, the cyclic hardening behavior of the gauge length can be described.

The TMT was conducted in the critically stressed volume in the cylindrical gauge length of the specimens with a length of 5 mm. Heating during the TMT was achieved by an induction coil, which was carefully adjusted to ensure a homogeneous temperature distribution within the gauge length. The temperature was measured using type K thermocouples applied at the center of the gauge length.

After the TMT, the specimens were cycled under the stress control with a sinusoidal waveform and a load ratio of R = −1 until failure or until the ultimate number of cycles $N_u$ was reached. For fatigue tests in the LCF and HCF regime, a frequency of 50 Hz was applied, and the ultimate number of cycles was $N_u = 10^7$. For the fatigue testing in the VHCF regime, a self-build ultrasonic piezoelectric fatigue testing device (components from Herrmann Ultraschalltechnik GmbH & Co. KG, Karlsbad, Germany) capable of cycling at a frequency of 20 kHz was used. A detailed description of the testing machine can be found in [24–26]. To prevent significant self-heating ($\Delta T < 40$ K) of the specimens, due to the high testing frequency, these tests were performed in the pulse-pause mode (resulting in an effective testing frequency $f_{effective} = 2.5$ kHz), and the specimens were cooled with a cold air gun (Vortec, Cincinnati, OH, USA). The ultimate number of cycles in the VHCF tests was $N_u = 10^9$. In both the HCF and VHCF regimes, the fatigue limit is considered as the maximum stress amplitude, which does not predominantly lead to failure after the ultimate number of cycles. The total number of fatigue tests in this study was not sufficient to determine the fatigue limit statistically.

Hardness values on longitudinal sections were measured using a Vickers-type micro-hardness tester Qness Q10 A+ (Qness GmbH, Salzburg, Austria) with an indentation load of 0.5 kg (HV 0.5).

### 2.3. Thermo-Mechanical Treatment

The TMT was conducted according to the procedure given in [18]. As a first step, the temperature at which the most pronounced DSA effects occur was determined by plastically cycling an HCF specimen at temperatures between 260 °C and 290 °C with a stress amplitude of 1500 MPa and a frequency of 1 Hz. It is assumed that the extent of DSA correlates to the amount of cyclic hardening. Thus, the ratio of qualitative plastic strain amplitude $\varepsilon_{a,p,quali}$ in the first and the 20th cycle was used to determine the temperature with the most pronounced DSA effects [18]. However, it is noticeable that in [18], the plastic strain amplitude has been considered. The mechanical loading of the TMT was then applied at this temperature, with gradually increasing stress amplitudes, according to [18]

and at a frequency of 1 Hz. For each stress amplitude, five cycles were applied, and the step between individual stress amplitudes was 100 MPa. The starting stress amplitude was 600 MPa, and the maximum stress amplitude was 1600 MPa.

### 2.4. Fracture Surface Analysis

The fracture surfaces of specimens were analyzed using SEM (Carl Zeiss AG, Oberkochen, Germany). The SEM images were used to measure the size of inclusions and the size of FGA. To determine the chemical composition of the non-metallic inclusions at the fracture surface, EDX (Thermo Fisher Scientific Inc., Waltham, MA, USA) was used.

## 3. Results and Discussion

### 3.1. Thermo-Mechanical Treatment

Figure 2 shows the cyclic hardening behavior of the material at various temperatures for a stress amplitude of 1500 MPa and a loading frequency of 1 Hz. The extent of cyclic hardening was evaluated using the ratio of the qualitative plastic strain amplitude in the first and 20th cycles. The maximum of this ratio and, thus, the highest extent of cyclic hardening was found at 265 °C, which is interpreted as the temperature of the maximum DSA for this strain rate.

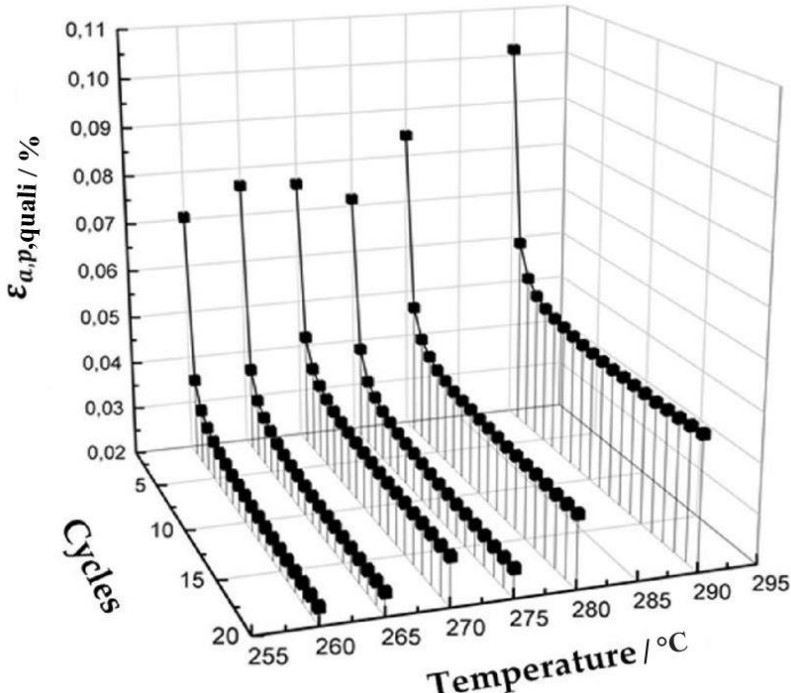

**Figure 2.** Development of the qualitative plastic strain amplitude $\varepsilon_{a,p,quali}$ at a constant stress amplitude of 1500 MPa and at 1 Hz for different temperatures. The ratio of $\varepsilon_{a,p,quali}$ in the first and 20th cycles was chosen to asses cyclic hardening.

After determining the temperature of the maximum DSA, a suitable range for the stress amplitudes at this temperature was identified to obtain an optimal strengthening effect. As described in [18], in order to strengthen the microstructure of the specimens gradually, increasing load amplitudes were applied. The first selected stress amplitude was 600 MPa since, for lower stress amplitudes, the plastic strain amplitude was negligible. The maximum stress amplitude was defined as the highest stress amplitude that leads to continuous cyclic hardening, therefore displaying decreasing plastic strain amplitudes within the five cycles of the loading step. Cyclic softening occurring at higher stress amplitudes is considered undesirable with possible damage induction [20]. Therefore, an increasing load amplitude at every fifth cycle by equidistant steps of 100 MPa from

600 MPa to 1600 MPa at 265 °C, with the frequency of 1 Hz, has been chosen for the TMT in this study.

Figure 3 shows a longitudinal section of a TMT-treated specimen with a hardness profile along the specimen axis. The macro-hardness profile was measured using Vickers hardness tests, with a 0.5 kg mass at the indenter and 30 equidistant positions with 0.75 mm distance between indentations. For positions 1 to 9, the hardness is significantly lower than for positions 10 to 30. The reduced hardness in the gauge length corresponds to the zone treated by the TMT. Consequently, the TMT-affected area can be estimated by the hardness measurement. The reduction in hardness is attributed to the thermal effect of the TMT. Since the TMT temperature of 265 °C is higher than the tempering temperature of 180 °C during the initial heat treatment, the TMT can be assumed to work as short-term tempering in this regard.

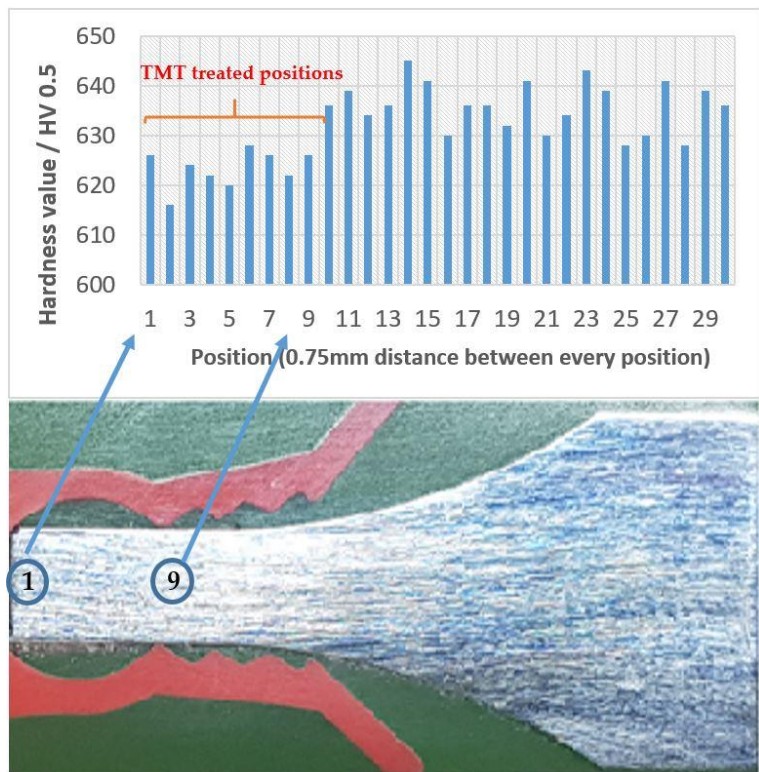

**Figure 3.** Influence of the TMT on the specimen hardness.

### 3.2. Lifetime Behavior

Figure 4 shows the S–N curve of the TMT-treated and the reference HT specimens tested on the servo-hydraulic testing setup in the LCF and HCF regimes. The amount of run-out specimens that reached $N_U = 10^7$ cycles without failure at a given stress amplitude is identified by the number beside the representative data point. For a stress amplitude of $\sigma_a \geq 900$ MPa, the fatigue life of the TMT-treated specimens is lower than for the reference HT specimens. In this lifetime regime, critical cracks initiate at the surface inclusions or the grooves on the surface. For $\sigma_a \leq 850$ MPa, TMT-treated specimens show higher lifetimes than non-treated specimens, and the critical crack initiated subsurface at non-metallic inclusions, which are typically of type AlCaO. Hence, the transition from the surface crack initiation to the subsurface crack initiation appears to also be the transition from lifetime decrease to lifetime increase due to the TMT. Both transitions occur in the stress amplitude range between 850 and 900 MPa. For $N_U = 10^7$, the fatigue limit of the TMT-treated specimens is approximately 50 MPa higher than for the reference HT specimens.

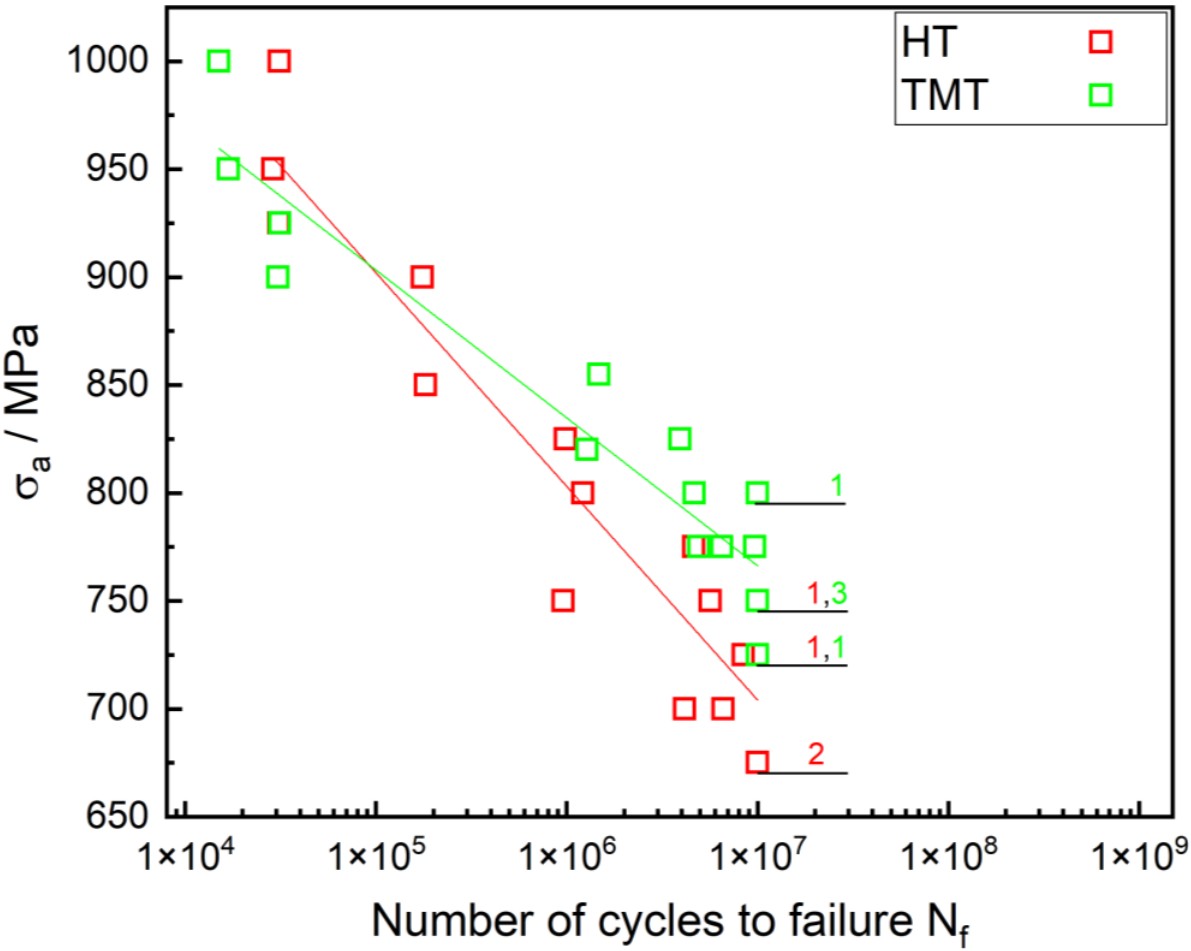

**Figure 4.** S–N curve for specimens with the TMT and HT in the HCF regime. The numbers besides the data points at $10^7$ cycles indicate the amount of run-out specimens.

The least square error approximation, represented by the colored lines in Figure 4, describes the data sets well (Coefficient of determination $R^2 = 0.90$ for HT specimens and $R^2 = 0.90$ for the TMT specimens) and confirms the impression that a TMT improves the fatigue properties in the HCF regime, while in the LCF regime, it has a negative effect on its lifetime. The reduced lifetime in the LCF regime may be attributed to the reduced hardness in the thermo-mechanical treated region (see Figure 3), which may lead to earlier surface crack initiation. Figure 5 compares the S–N curves for the TMT and the reference HT specimens in the VHCF regime. With the exception of one specimen failing at a surface inclusion, all specimens tested with the ultrasonic testing setup failed at more than $4 \times 10^5$ cycles and showed crack initiation at non-metallic inclusions of type AlCaO inside the volume. As in Figure 4, the amount of runout specimens at a given stress amplitude is represented by the respective colored numbers beside the data points. Although the scattering of lifetimes is much higher than in the HCF regime, which is reflected by a rather low coefficient of determination $R^2$ of 0.42 for the TMT specimens and 0.39 for the HT specimens, there is a clear trend that both VHCF fatigue life and fatigue strength for $N_U = 10^9$ are increased by a TMT.

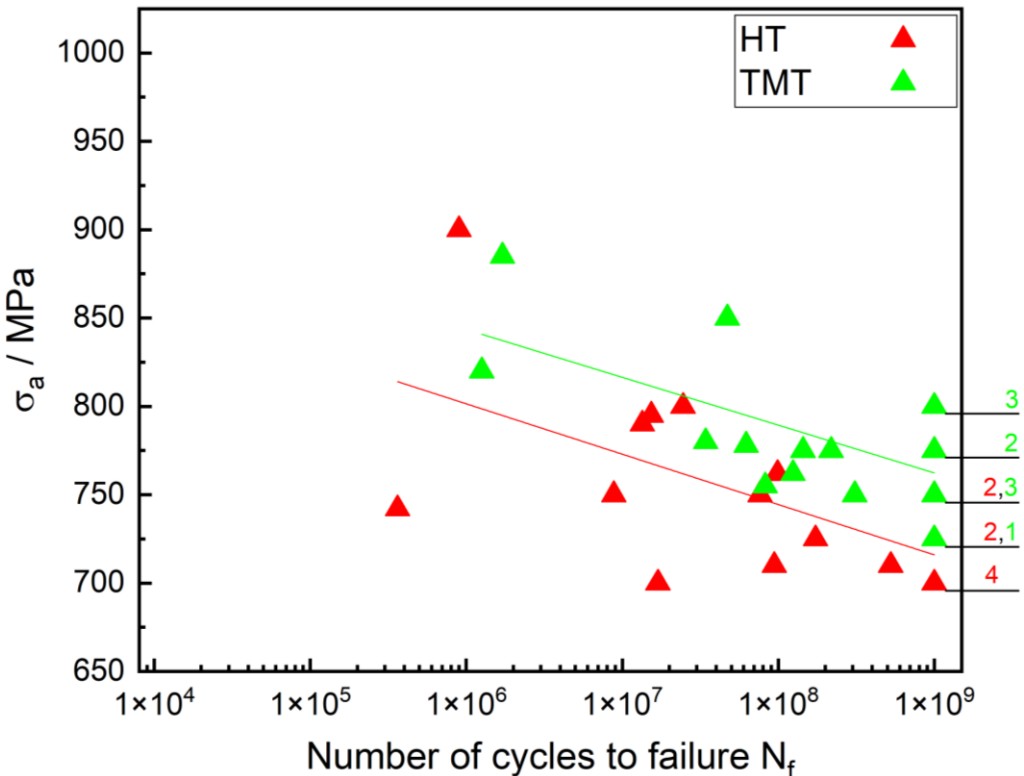

**Figure 5.** S–N curve for specimens with the TMT and HT in the VHCF regime. The numbers besides the data points at $10^9$ cycles indicate the amount of run-out specimens.

Hence, for the HCF and VHCF regimes, the TMT has the expected positive effect on fatigue lifetime and fatigue strength. Both can be explained by the cyclic plastic deformation at the temperature of maximum DSA, where the dislocation velocity is approximately the same as the diffusion rate of carbon atoms. When diffusing solute carbon atoms are gathered in the dilatation zone of mobile edge dislocations, they can be locked, which leads to a stabilized dislocation structure. It is assumed that the effect is most pronounced around inclusions due to the stress concentration around them. Consequently, volume crack initiation at inclusions due to cyclic dislocation movement is delayed or prevented. Since crack initiation at inclusions inside the volume is the dominating damage mechanism in the HCF and VHCF regimes, increased fatigue lifetimes and a higher fatigue strength occur. For crack initiation at inclusions, the positive effect of dislocation locking due to the TMT does apparently outweigh the negative thermal effect of the TMT on hardness. In the LCF regime, the stress amplitudes are presumably high enough to unlock dislocations easily; thus, the positive effect of the TMT is lost, and lifetimes are reduced due to the lower hardness.

Comparing the lifetime results for servo-hydraulic and ultrasonic testing, one can see that the stress amplitudes at which runouts occur are similar. Since the VHCF fatigue strength should be lower than the HCF fatigue strength, this indicates a possible influence of the testing frequency. The strain rate in the VHCF tests is significantly higher than in the HCF tests, which may result in an increased fatigue strength for low- and medium-strength steels [27]. However, numerous other investigations found no significant influence of the testing frequency [22,28,29]. From our investigations, there is a distinct positive effect of a TMT treatment on HCF as well as VHCF lifetimes, and fatigue strength can be derived. However, it should be noted that, from a statistical point of view, more experiments should be conducted for the individual variants. Furthermore, test series on the specimens with the applied mechanical treatment at room temperature, as well as on specimens with only a thermal treatment at 265 °C, should be carried out in order to better

differentiate between the pure mechanical effect, the pure thermal effects and the combined thermomechanical effects.

### 3.3. Fractography and Damage Analysis

For a better understanding of the mechanisms responsible for the increase in fatigue strength due to the TMT, all fracture surfaces have been analyzed using SEM imaging. Figure 6 shows the representative crack initiation sites at the surface (left) and inside the volume at a non-metallic inclusion surrounded by a fisheye (right). The qualitative appearance of crack initiation sites was comparable for both the TMT and reference HT specimens. Since this study is focused on the fatigue behavior in the HCF and VHCF regimes, only crack initiation due to internal inclusions was considered.

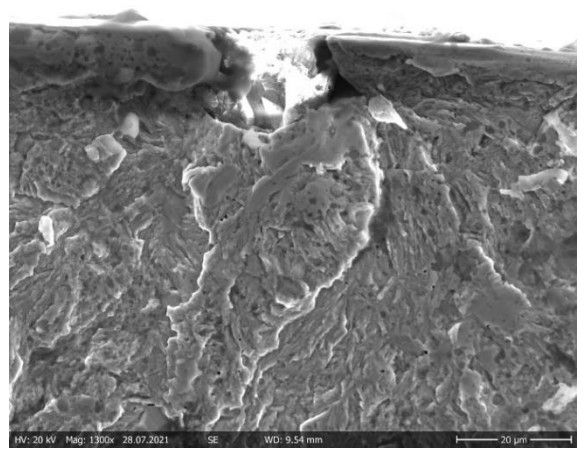 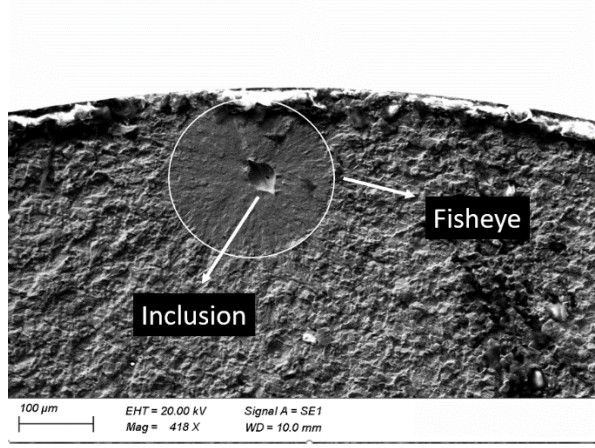

(**a**) HT $\sigma_a$ = 900 MPa, $N_f$ = 175,162 cycles      (**b**) with TMT $\sigma_a$ = 775 MPa, $N_f$ = 4,932,070 cycles

**Figure 6.** SEM images of two fracture surfaces: (**a**) crack initiation at a surface groove; (**b**) crack initiation at a critical inclusion inside the volume as well as the fisheye formation.

The critical inclusions of both the TMT-treated and HT specimens have been analyzed with regard to their chemical composition, and it has been found that for all analyzed specimens, the inclusions containing aluminum and calcium, in particular AlCaO, are the crack initiation site since they are typically much larger than titanium nitride inclusions and are, therefore, the most deleterious [30].

With the measured area of the crack-initiating inclusions, the maximum stress intensity factor for sub-surface inclusions is derived by Equation (1) [1]:

$$K_{max,Inc} = 0.5 \cdot \sigma_{max} \cdot \sqrt{\pi \cdot \sqrt{area_{inc}}} \tag{1}$$

In Figure 7, the maximum stress intensity factor of the non-metallic inclusions ($K_{max,Inc}$) in the HCF and VHCF regimes for both the TMT and HT specimens are plotted versus the lifetime. An expected general trend of decreasing $K_{max,Inc}$ values with increasing lifetime is observed.

In the HCF regime, for comparable lifetimes, the values of $K_{max,Inc}$ are higher for the TMT specimens than for the reference HT specimens. This suggests that the damage tolerance of the surrounding microstructure was increased by the TMT, thus resulting in higher lifetimes in this regime (see Figure 4). Comparing $K_{max,Inc}$ values of the TMT and the reference HT specimens in the VHCF regime, the differences are not as pronounced as in the HCF regime. However, particularly for $K_{max,Inc}$ values between 3.5 and 4.25 MPa·m$^{1/2}$, the TMT specimens show higher average lifetimes; the results confirm the strengthening effect on the microstructure around inclusions due to the TMT.

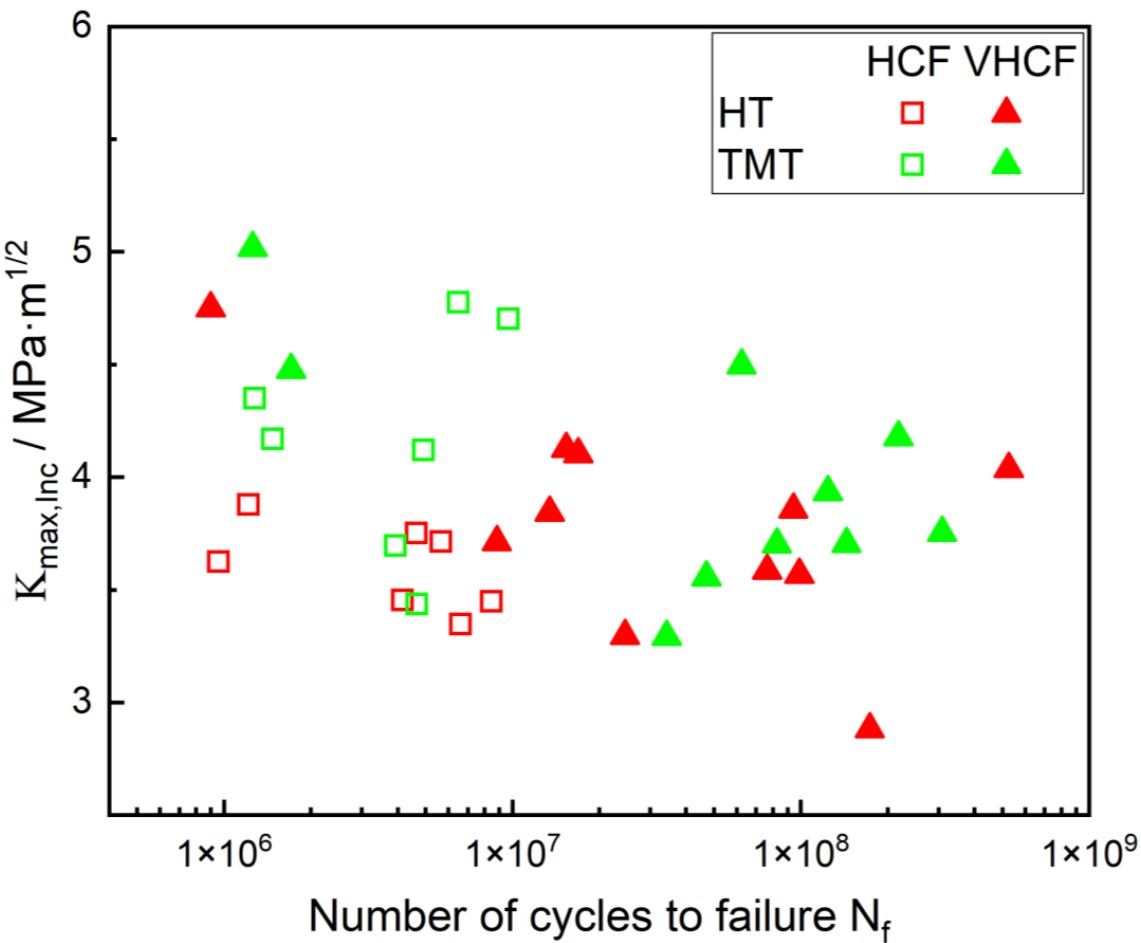

**Figure 7.** Maximum stress intensity factors $K_{max,Inc}$ at internal critical non-metallic inclusions over a lifetime.

Figure 8 illustrates a representative sub-surface crack initiation with an FGA formation. In the VHCF regime, a pronounced FGA is found in all cases, while in the HCF regime, most of the HT specimens and all of the TMT specimens showed an FGA formation at the inclusions. Additionally, Figure 8 displays the area of the inclusion (blue) used to calculate $K_{max,Inc}$ and the area of the FGA (red) used to calculate the stress intensity factor at the border of the FGA ($K_{max,FGA}$) in analogy to Equation (1), using area$_{FGA}$ instead of area$_{Inc}$.

FGA formation around the deleterious inclusion was observed for both the HT and TMT specimens for $N_f \geq 9 \times 10^5$ cycles. By means of SEM observation, no difference between the FGAs of the HT and TMT specimens could be found. FGA formation is only observed in the HCF and VHCF regimes when $K_{max,Inc} \leq 5$ MPa·m$^{1/2}$ (compare Figure 7), which is in line with several findings of other investigations on similar high-strength steels [31–33]. Figure 9 depicts the stress intensity factors of the observed FGAs $K_{max,FGA}$ over the lifetime of the HT and TMT specimens. The $K_{max,FGA}$ values show a small range of $\approx 4.5$–6 MPa·m$^{1/2}$. The values agree with other investigations on this topic and correspond with the threshold value for the long crack propagation observed in high-strength steels [6,31,34–36]. The fact that there is no obvious difference of $K_{max,FGA}$ between the HT and TMT leads us to the conclusion that the TMT does not influence the transition from FGA formation to the long-crack growth phase. Hence, although the TMT cannot prevent FGA formation, the results suggest that it is delayed. Therefore, the main benefit from the TMT is the retardation of the crack initiation and early crack formation directly at the inclusion. However, for a better understanding, further research is necessary.

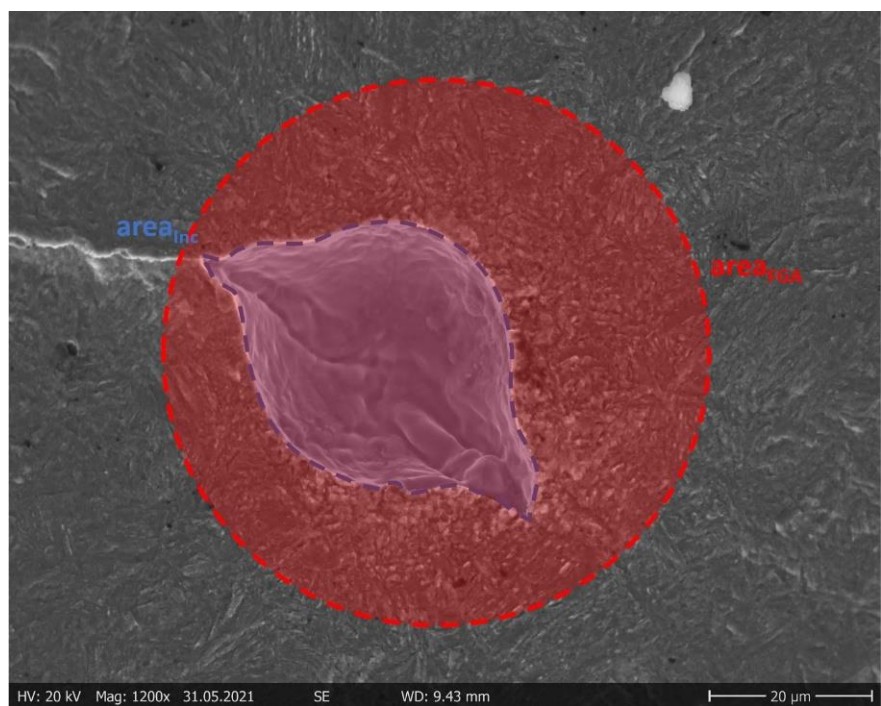

**Figure 8.** Crack initiation on the internal inclusion with a pronounced FGA formation (TMT $\sigma_a$ = 755 MPa, $N_f$ = 82,557,794 cycles) and the areas used for calculation of $K_{max,Inc}$ (blue-marked are of the inclusion) and $K_{max,FGA}$ (red-marked are of the area of inclusion plus FGA).

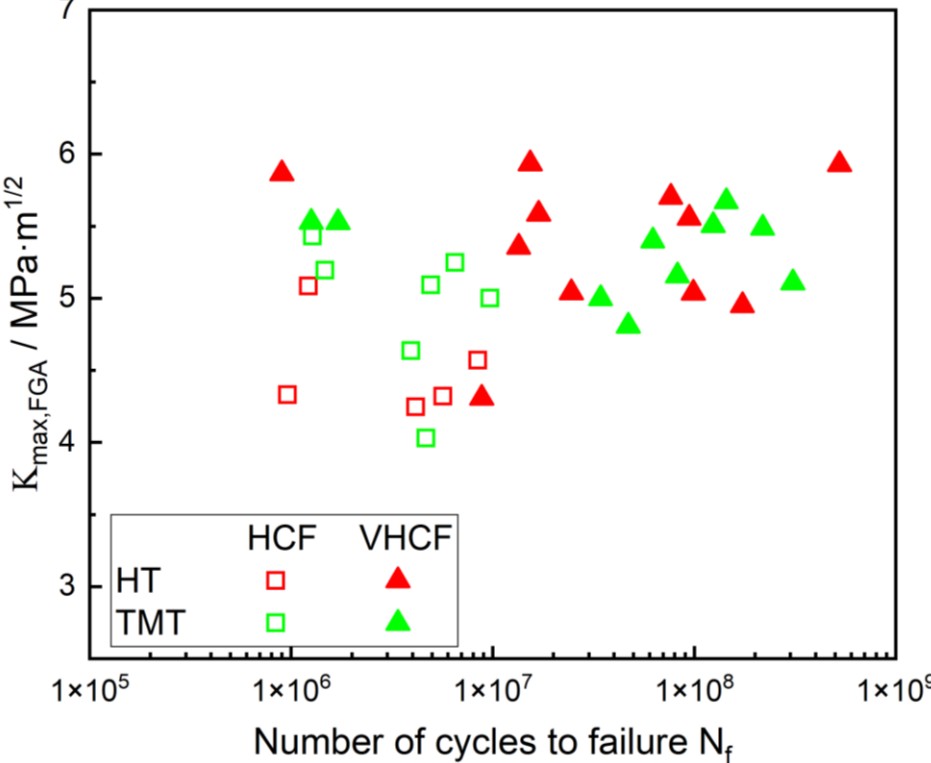

**Figure 9.** Maximum stress intensity factor at the border of FGA $K_{max,FGA}$ for the HT and TMT specimens.

## 4. Conclusions

In this study, a thermo-mechanical treatment (TMT) was optimized to increase the fatigue limit of quenched and tempered steel SAE4140 (42CrMo4). Fatigue experiments were conducted on the TMT specimens and on the non-treated reference specimens in the LCF, HCF and VHCF regimes. The results can be summarized as follows:

1. A suitable TMT, with the optimum temperature of 265 °C and a gradually increasing stress amplitude of 600–1600 MPa, was found and applied.
2. The applied TMT increases both the fatigue lifetime and the fatigue strength in the HCF and VHCF regimes, while in the LCF regime, it leads to a reduced lifetime. The lower lifetime in the LCF regime may be attributed to the lower hardness due to the TMT, which acts like a short-term-tempering. Consequently, in the HCF and VHCF regimes, the TMT overcompensates the negative effect of the lower hardness.
3. The critical inclusion sizes, leading to the crack initiation in both the TMT and HT specimens, have been measured, and it has been found that the maximum stress intensity factors of inclusions for the TMT specimens are higher than those for the reference specimens in the case of failure in the HCF and VHCF regimes. The positive effect of the applied TMT, resulting in the observed increased fatigue strength and fatigue lifetime, can be, thus, attributed to a stabilization of the microstructure around the critical inclusions due to cyclic plastic deformation at the temperature of maximum dynamic strain aging during the TMT.
4. While the TMT was able to increase the fatigue strength in the HCF as well as in the VHCF regime, it was unable to prevent the characteristic FGA formation, resulting in a failure in the VHCF regime, and it does not modify the transition from FGA formation to long crack growth at the border of the FGA.

**Author Contributions:** Conceptualization, K.-H.L. and E.K.; investigation, A.K. and J.S.; writing—original draft preparation, A.K. and J.S.; writing—review and editing, S.G. and K.-H.L.; visualization, A.K. and J.S.; supervision, S.G. and E.K.; funding acquisition, K.-H.L. and E.K. All authors have read and agreed to the published version of the manuscript.

**Funding:** This research project was funded by the Deutsche Forschungsgemeinschaft (DFG, German Research Foundation)—Projektnummer 408139037.

**Data Availability Statement:** Not applicable.

**Conflicts of Interest:** The authors declare no conflict of interest.

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
