# Peer review of "Influence of a Thermo-Mechanical Treatment on the Fatigue Lifetime and Crack Initiation Behavior of a Quenched and Tempered Steel"

_metals, doi:10.3390/met12020204_

Round 1

Reviewer 1 Report

This paper presents the effect of a thermo-mechanical treatment (TMT) on the fatigue lifetime of a high strength steel SAE4140H (German designation: 42CrMo4).

The method to choose the best TMT was already published in reference [18-19]. So, it is not innovative in this paper, but it was successfully applied for another high strength steel sensitive to DSA. Additional analyses have been carried on the maximum stress intensity factor at the inclusions and the FGA.

The paper is interesting and well written.

Nevertheless, I recommend the paper for publication after taking into account the revisions listed below in order to clarify some issues and improve the manuscript.

  1. Authors make an excessive use of thesis references [20 - 24 -25] which are not available in English. Could the authors find other references in scientific journals?
  2. Why is there a thread at each end of the VHCF-specimen Figure 1? Indeed the load ratio is R = -1, so one end should be free.
  3. It seems according to Figure 2 that εa,p,quali is lower at 260 °C than 265 °C. So, why 265 °C is considered as the temperature of maximum DSA? Moreover, how is εa,p,quali at lower temperature (200 to 255 °C for example)?
  4. p. 5 - l. 155: authors give a stress threshold of 875 MPa, but why don't the authors take the intersection of the lines drawn Figure 4 as stress threshold to distinguish the lifetime behavior between HT and TMT specimens?
  5. Write a real multiplication sign in Equation 1 and everywhere in the text.
  6. p. 6 - l. 185: a coma between "dislocation" and "they" would be suitable.
  7. Do the authors try to apply the same mechanical treatment (increasing load amplitude at every fifth cycle by equidistant steps of 100 MPa from 600 MPa to 1600 MPa) at room temperature to ensure the effect of the TMT, and therefore prove the fatigue limit enhancement by DSA and not only classical strain hardening?
  8. Reviewer cannot find Equation 2 in reference [1]. Could authors check this reference?
    Moreover, this equation differs somewhat from the classical relation for a circular crack with radius a in an infinite bulk subjected to a homogeneous orthogonal stress σ: KI = 2/π*σ*sqrt(πa). Can the authors explain this?
    Authors should also precise what is areainc and areaFGA, even if it seems trivial.
  9. p. 8 - l. 230-231: Authors state: "In the VHCF regime a similar trend of higher Kmax,Inc values for TMT specimens than for reference HT specimens can be observed". It is not at all obvious on Figure 7, also given the measurement uncertainties (not provided here). Authors can argue or change this assertion?
  10. Figure 8: through SEM observations, is it possible to differentiate FGA for HT and TMT specimens?
  11. By computing the Kmax,FGA with Equation 2 and considering the surface of the FGA area drawn in Figure 8 (FGA diameter ≈ 57 µm), the reviewer finds Kmax,FGA = 4,75 MPa.m1/2 for σa = 755 MPa. Consequently, there is an inconsistency with the results in Figure 9. The authors must check their results.
  12. p. 9 - l. 246 and p. 10 - l. 251: add a dot at the unit "MPa.m1/2"
  13. p. 10 - l. 253-254: Authors state: "comparable Kmax,FGA values result in higher lifetimes for TMT specimens". It is very difficult to deduce this conclusion from Figure 9. Authors can argue or change this assertion?

Author Response

Dear Reviewer,

First, thank you very much for your effort. We really appreciate your constructive feedback. Therefore, we have considered your comments. Please find our changes of the paper and some more feedback below. Thank you once again for investing your valuable time to review the paper!

Best regards

Amin Khayatzadeh, Jan Sippel, Stefan Guth, Karl-Heinz Lang and Eberhard Kerscher

Response to Reviewer 1 Comments

Comment 1: Authors make an excessive use of thesis references [20 - 24 -25] which are not available in English. Could the authors find other references in scientific journals?

Response 1:

English references have been added. The dissertations are to be understood merely as additional supplementary references in which the experiments are described in an extensive manner.

Comment 2: Why is there a thread at each end of the VHCF-specimen Figure 1? Indeed, the load ratio is R = -1, so one end should be free.

 Response 2:

You are obviously right, that for the case of the load ratio being R = -1 the second thread is not needed. However, the specimens were designed in such a way that potential tests with different load ratios as well as tests in vacuum can be carried out. For this purpose, another Mason’s horn must be mounted at the end of the specimen to realise the mounting. The testing equipment is calibrated with the additional mason-horn that was also mounted during the tests shown here.

Comment 3: It seems according to Figure 2 that εa,p,quali is lower at 260 °C than 265 °C. So, why 265 °C is considered as the temperature of maximum DSA? Moreover, how is εa,p,quali at lower temperature (200 to 255 °C for example)?

Response 3:

The criterion chosen to describe the cyclic hardening was not the absolute values of εa,p,quali after 20 cycles, but the ratio of εa,p,quali in the first and the 20th cycle, which is in fact greatest for T = 265 °C. The text has been modified and a phrase in the description of Fig. 2 has been added to clarify this. For lower temperatures than 260 °C, the there was less pronounced cyclic hardening. 

Comment 4: p. 5 - l. 155: authors give a stress threshold of 875 MPa, but why don't the authors take the intersection of the lines drawn Figure 4 as stress threshold to distinguish the lifetime behavior between HT and TMT specimens?

Response 4:

After critically re-examining all the data, we had to modify the diagram, as we cannot make a clear distinction between surface failure and internal failure in terms of numbers of cycles to failure in the HCF regime, due to the marked data point corresponding to a surface defect in Figure 4 and the data point has been removed from Figure 7 since only internal inclusions are supposed to be considered/discussed in Figure 7.

In addition, a near-surface inclusion has been identified and reconsidered. Since neither FGA formation nor fisheye formation is observed at this inclusion, it is considered as a surface inclusion, now. Consequently, there is a change in Figure 5.

While carefully re-examining all data points (For the marked data point of an internal inclusion KmaxInc, as well as KmaxFGA was falsely calculated with 0.65) and no longer including the two surface inclusions discussed above Figure 7 have to be changed to:

Comment 5: Write a real multiplication sign in Equation 1 and everywhere in the text.

 Response 5:

Thank you for pointing this out, it has of course been replaced in all cases.

Comment 6: p. 6 - l. 185: a coma between "dislocation" and "they" would be suitable.

 Response 6:

Again, thanks for correction, the appropriate comma has been inserted.

Comment 7: Do the authors try to apply the same mechanical treatment (increasing load amplitude at every fifth cycle by equidistant steps of 100 MPa from 600 MPa to 1600 MPa) at room temperature to ensure the effect of the TMT, and therefore prove the fatigue limit enhancement by DSA and not only classical strain hardening?

 Response 7:

A very good remark. Indeed tests with purely mechanical stress, which corresponds to the mechanical stress carried out during TMT, as well as tests with purely thermal exposure analogous to TMT are planned. Some discussion and outlook on this was added at the end of paragraph 3.1

Comment 8: Reviewer cannot find Equation 2 in reference [1]. Could authors check this reference?

Moreover, this equation differs somewhat from the classical relation for a circular crack with radius a in an infinite bulk subjected to a homogeneous orthogonal stress σ: KI = 2/π*σ*sqrt(πa). Can the authors explain this?

Authors should also precise what is areainc and areaFGA, even if it seems trivial.

Response 8:

The Equation is given on page 293 as Equation (5) in reference [1] and describes maximum values of stress intensity factor for internal inclusions. This equation is commonly used in literature to describe Kmax of internal inclusions.

The distinction between areaInc and areaFGA is now explained properly by the schematic representation in Figure 8.

Comment 9: p. 8 - l. 230-231: Authors state: "In the VHCF regime a similar trend of higher Kmax,Inc values for TMT specimens than for reference HT specimens can be observed". It is not at all obvious on Figure 7, also given the measurement uncertainties (not provided here).  Authors can argue or change this assertion?

Response 9:

Obviously, the evaluation in the VHCF regime is more difficult due to the higher scattering in lifetime. However, with comparable stress intensity factors (in the range of 3.5 to 4.25), four of the five longest lifetimes are attributable to TMT specimens. Additionally, at these stress intensity factors, four HT specimens are observed to fail a full decade earlier. The mentioned paragraph was changed in order give a more precise statement.

Comment 10: Figure 8: through SEM observations, is it possible to differentiate FGA for HT and TMT specimens?

Response 10:

By means of SEM, no differences between FGAs of HT and TMT specimens could be found. An according phrase was added. A more detailed analyses using TEM lamellae may be useful for future investigations.

Comment 11: By computing the Kmax,FGA with Equation 2 and considering the surface of the FGA area drawn in Figure 8 (FGA diameter ≈ 57  µm), the reviewer finds Kmax,FGA = 4,75 MPa.m1/2 for σa = 755 MPa. Consequently, there is an inconsistency with the results in Figure 9. The authors must check their results.

Response 11:

Thank you for pointing this out, Figure 9 did not show the area of the FGA determined for the calculation (taking both fracture surfaces into account), but only used a qualitative schematic area. The figure has been adjusted to represent the actual area used.

 Comment 12: p. 9 - l. 246 and p. 10 - l. 251: add a dot at the unit "MPa.m1/2"

 Response 12:

Thanks for the comment: MPam1/2 has been replaced by MPa·m1/2 in text as well as in the Diagrams.

Comment 13: p. 10 - l. 253-254: Authors state: "comparable Kmax,FGA values result in higher lifetimes for TMT specimens". It is very difficult to deduce this conclusion from Figure 9. Authors can argue or change this assertion?

 Response 13:

After discussing the Issue, we agree with your observation. Therefore, the interpretation has been changed.

Reviewer 2 Report

Metals-1524948

This paper is an experimental investigation regarding the influence of a thermo-mechanical treatment (TMT) on the high cycle and very high cycle fatigue of a 42 CrMo4 steel. The TMT was performed in the range of temperature where dynamic strain ageing occurs. The proper temperature of the TMT treatment was singled out by plastically cycling a specimen at different temperatures to investigate the amount of cyclic hardening. The effect of the TMT was interpreted on the basis of the maximum stress intensity factor at the killer inclusion versus the number of fatigue cycles, revealing the higher fatigue strength of the TMT specimens as compared with the HT specimens.

In my opinion this paper reports on an interesting investigation, it is well organised and clearly written therefore it is acceptable for publication. I have just a couple of comments:

  • Section 2.3: please clarify what is meant by “qualitative plastic strain”
  • (1): it is not clear to me why the Authors invoke Eq. (1) to justify the lower fatigue strength of TMT specimens in the low-cycle fatigue regime. Eq. (1) describes the mechanical threshold of short cracks/defects therefore it seems not appropriate to describe low cycle fatigue damage.

Author Response

Dear Reviewer,

First, thank you very much for your effort. We really appreciate your constructive feedback. Therefore, we have considered your comments. Please find our changes of the paper and some more feedback below. Thank you once again for investing your valuable time to review the paper!

Best regards

Amin Khayatzadeh, Jan Sippel, Stefan Guth, Karl-Heinz Lang and Eberhard Kerscher

Response to Reviewer 2 Comments

 Comment 1: •            Section 2.3: please clarify what is meant by “qualitative plastic strain”.

 Response 1:

The term is explained in section 2.2.:

“The prolongation of HCF-specimens in experiments to find the temperature of maximum DSA effects was measured with an extensometer that was applied outside of the specimen’s gauge length using alumina rods. The measuring length of the extensometer was 30 mm. With the measured prolongation of the specimen, the actual strain within the cylindrical gauge length cannot be determined, since the radii towards the specimen shoulders are within the measuring length. However, with the applied setup, a qualitative strain in the measuring length of the extensometer can be determined, which is in turn used to obtain stress-strain hysteresis loops and an according qualitative plastic strain amplitude εa,p,quali. It is assumed that with this qualitative plastic strain amplitude, the cyclic hardening behavior of the gauge length can be described.”

We modified the paragraph slightly in order to make it easier to find.

Comment 2: It is not clear to me why the Authors invoke Eq. (1) to justify the lower fatigue strength of TMT specimens in the low-cycle fatigue regime. Eq. (1) describes the mechanical threshold of short cracks/defects therefore it seems not appropriate to describe low cycle fatigue damage.

 Response 2:

We assumed that with a lower value for ΔKth, the whole crack propagation curve is shifted leading to generally higher crack propagation rates and thus lower LCF lives. We agree, that this assumption is not necessarily true. The passage was modified accordingly.

Reviewer 3 Report

  • Fatigue limit are not for avery materiasl. as is written l.24-25. It is valid for steels, not for example for aluminium alloy
  • l.52/53 and 60 therea are written 2 times (German designations: 42crMo4). It is important inforamtion, but authors should write it 1 times
  • l.144/145 it is written "0.5 kg load". It is mass not load (force)
  • generally in paper aouthors write abot fatigue limit. Bot is rather amplitude in .... number of cycles. This parameter is known as rather as knee point (in German literaure - see Sonsino and others)
  • Figures 5, 7, 9 - in axis X please add - number of cycles
  • in all text it should be written MPa·m (not MPam)

Author Response

Dear Reviewer,

First, thank you very much for your effort. We really appreciate your constructive feedback. Therefore, we have considered your comments. Please find our changes of the paper and some more feedback below. Thank you once again for investing your valuable time to review the paper!

Best regards

Amin Khayatzadeh, Jan Sippel, Stefan Guth, Karl-Heinz Lang and Eberhard Kerscher

Round 2

Reviewer 1 Report

Authors honestly responded to all the comments and questionings.

I recommend now the paper for publication after minor corrections, where I would advise the authors to take into consideration these three last comments:

  • Comment 4: authors have not really answered to the question about the choice of the stress amplitude 875 MPa as stress threshold.
  • Comment 7: alone, a thermal treatment at 265 °C should have no effect on fatigue behaviour, because nothing happens at this temperature (except perhaps a slight stress relaxation, but no aging). So, authors should remove the case for the thermal effect.
  • Authors should give the exact composition of the steel (especially in terms of carbon and nitrogen content) for further investigations on DSA effect and comparison with other materials.

Author Response

Dear Reviewers,

Thank you very much again for your valuable further feedback, which we appreciate very much. Therefore, we have considered your comments. Please find our changes of the paper and some more feedback below. Thank you once again for investing your valuable time to review the paper!

Best regards

Amin Khayatzadeh, Jan Sippel, Stefan Guth, Karl-Heinz Lang and Eberhard Kerscher

Response to Reviewer 1 Comments

  • Comment 4: authors have not really answered to the question about the choice of the stress amplitude 875 MPa as stress threshold.

Response 4: You are right that the threshold of 875 MPa has been chosen somewhat arbitrarily. The intersection of the regression lines lies above 900 MPa where we already see longer lifetimes for HT-specimens: Hence, we think that the threshold or better transition should be at a lower stress amplitude. Since we cannot determine the transition stress amplitude more precisely with the available data, we modified the statements in the text now stating that it appears to be between 850 and 900 MPa.

  • Comment 7: alone, a thermal treatment at 265 °C should have no effect on fatigue behaviour, because nothing happens at this temperature (except perhaps a slight stress relaxation, but no aging). So, authors should remove the case for the thermal effect.

Response 7: Actually, we have already conducted some tests on purely thermally treated specimens. Until now it seems that the lifetime behaviour is significantly different, when compared with initial state “HT”-specimens: Lifetimes in LCF and HCF regimes as well as fatigue strength seem to be lower. The reason for this might be residual stress relaxation as well as hardness decrease due to annealing effect (annealing temperature during heat treatment was 180 °C, so the treatment temperature of 265 °C could lead to additional annealing effects). The results have not been published so far; however, we intend to do so. Therefore, we would prefer to mention the influence of a thermal treatment in the outlook of this manuscript.

  • Authors should give the exact composition of the steel (especially in terms of carbon and nitrogen content) for further investigations on DSA effect and comparison with other materials.

Response: We added the chemical composition to section 2.1
